# Exploring perceptions of gender roles amongst sexually active adolescents in rural KwaZulu-Natal, South Africa

Brett Marshall[1]☯, Celia Mehou-Loko[2]☯, Sindisiwe Mazibuko[1], Makhosazana Madladla[1], Lucia Knight[3,4], Hilton Humphries🄳[5,6]*

1 Centre for the AIDS Programme of Research in South Africa (CAPRISA), Nelson R Mandela School of Medicine, University of KwaZulu-Natal, Durban, South Africa, 2 Department of Pathology, Institute of Infectious Disease and Molecular Medicine (IDM) and Division of Medical Virology, University of Cape Town, Cape Town, South Africa, 3 Faculty of Health Sciences, Division of Social and Behavioural Sciences, School of Public Health, University of Cape Town, Cape Town, South Africa, 4 School of Public Health, Community and Health Sciences, University of the Western Cape, Bellville, South Africa, 5 Centre for Community Based Research, Human Sciences Research Council, Pietermaritzburg, South Africa, 6 Department of Psychology, School of Applied Human Sciences, University of KwaZulu-Natal, Pietermaritzburg, South Africa

☯ These authors contributed equally to this work.
* hhumphries@hsrc.ac.za

**Data Availability Statement:** All relevant data is in the manuscript.

## Abstract

Traditional gender and social norms reinforce asymmetrical power relations, increase the risk of experiencing gender-based violence and mediate poor engagement with sexual and reproductive health services. This study explored gender norms and expectations amongst cisgender adolescents in rural KwaZulu-Natal, South Africa. A purposive sample of 29 adolescents aged 16–19 years old were enrolled as part of a longitudinal qualitative study. The current analysis reports on the first round of in-depth interviews, which focused on the role of men and women in their community. A theoretically informed thematic analysis identified three broad themes: 1) *Adolescent interpretation and understanding of gender identity*, 2) *Gendered essentialism* and *Gender roles* (two sub-themes: Young men: Power through providing, and Young women: The domestication process which highlighted that gender roles were defined by being the provider for men, and the successful fulfilment of traditional domestic behaviours amongst women), 3) *Gender and fertility* highlighted how participants highly valued fertility as affirming of manhood/womanhood. These norms reinforce gender roles that maintain asymmetrical power relations, carrying them over into adulthood. The subtle social pressure to prove fertility could have unintended consequences for driving teenage pregnancy. Structural, gender-based interventions emphasising positive gender-role development in early childhood are needed.

## Introduction

Gender inequalities are a key driver of negative sexual and reproductive health (SRH) outcomes, such as HIV, sexually transmitted infections (STI) and unplanned and unwanted

**Funding:** Financial Disclosure Statement. This work was supported by the South Africa National Research Foundation's Thuthuka Funding under Grant [number TTK200403511178]. There was no additional external funding received for this study. The funders had no role in study design, data collection and analysis, decision to publish, or preparation of the manuscript.

**Competing interests:** The authors have no conflicts of interest to declare.

pregnancies [1]. In sub-Saharan Africa (SSA), women accounted for 63% of new HIV infections in 2021 [1]. Adolescent girls and young women (AGYW) aged 15–24 years in the region are at highest risk, being three times more likely to acquire HIV than their male counterparts and have seen a slower decline in new HIV infections over the last decade [1,2]. South Africa has the world's largest HIV pandemic and AGYW have a four times higher risk of acquiring HIV than their male peers [3] and 16% of AGYW aged 15–19 years old have begun childbearing [4].

Gender inequalities, unequal power dynamics, socially sanctioned gender roles and harmful gender and adolescent sexuality norms are important mediators of increased vulnerability to negative SRH outcomes amongst AGYW [1,5]. Studies looking at gender norms (social norms defining acceptable and appropriate actions for women and men in a given group or society) indicate that unequal relational power between men and women, coupled with traditional beliefs about masculine and feminine gender norms, enhance risk of negative SRH outcomes [6–10]. Unequal power dynamics and harmful gender norms also have a negative impact on women's ability to make independent SRH decisions, increasing their risk of sexual and gender-based or intimate partner violence, and promotes negative gendered expectations (i.e., girl children dropping out of school to provide support at home) that keep women dependent on their partners/families and perpetuate gender inequality [1,11,12].

Social and cultural contexts will vary, so understanding how gender norms and expectations may impact adolescent risk-taking behaviours and sexual decision-making, especially in high-risk settings is critical [13–16]. Gender norms are particularly important during adolescence as gender roles and power imbalances become more pronounced, alongside sexual exploration and the initiation of sexual activity [1,17]. Indeed, research indicates that power becomes essential in the formation of masculine gender roles and therefore young men may assert this power in their relationships with young women [8,18]. This extends to the sexual realm where gender norms such as virility and power become intrinsic to manhood and modesty and submission become intrinsic to womanhood [9,13,19]. Gender norms premised on power and submission reinforce asymmetrical power relations and may cause poor sexual negotiation dynamics and unilateral decision-making by men [18]. Gender norms may further reinforce poor SRH seeking behaviours amongst young people as displays of weakness or as confirmation of actual or planned sexual engagement, especially in contexts with conservative norms around adolescent sexuality. Importantly, norms entrenched during adolescence may subsequently influence adulthood [1].

In the South African context, research shows that gender and social norms often overlap and may influence how adolescents navigate their sexual lives and affirm their gender success [6,15,16,20]. These gender norms may indirectly impact sexual risk behaviour, by conflating demonstrated female fertility with successful womanhood and virility and relational dominance with successful manhood [16]. More research is needed to explore how gender norms may affect sexual decision-making and SRH outcomes for adolescents and young people. This paper therefore explores gender norms and expectations amongst cis-male and cis-female adolescents in a single high-risk, HIV-endemic rural setting and how this may affect SRH outcomes and decision-making.

## Materials and methods

### Study community

The study took place within the rural community of Vulindlela. Vulindlela is located 150km west of Durban in the uMgungundlovu District Municipality, in central KwaZulu-Natal. Vulindlela is comprised of a population of just under 160,000 predominantly isiZulu speaking

people and is characterised by high community HIV prevalence [21], as well as low rates of marriages, high rates of STIs, high secondary school dropout rates and high teenage pregnancy rates [21,22].

## Procedure

The parent study uses a longitudinal qualitative design that involved enrolling male and female adolescents to participate in six in-depth interviews conducted every two months for a year (June 2021 to Feb 2023), complete a diary and participate in a quarterly photovoice activity. This analysis uses data from the first round of in-depth interviews (June to December 2021), which explored the roles and expectations of men and women and the gender and social norms that characterise sex and sexuality for men and women within this community.

Purposive sampling was used to identify participants living in Vulindlela aged between 15–19 years old who self-reported being sexually active (having had sex in the last month). *This age range was selected because it included those in mid-adolescence and are likely have had their sexual debut more recently, reducing potential issues with the recall of early sexual behaviour. This age group is also likely to be in similar orts of sexual relationships (i.e., short term) compared to older adolescent groups where longer-term and formal relationships are more likely to occur and drivers of sexual behaviour may change.* Recruitment was done through outreach activities in the community and through existing adolescent networks. Snowball sampling was also used to identify potential participants by asking enrolled participants if they had peers who would be interested in participating. Participants were asked to provide potential volunteers with information about the study and a number where they could send *a please call me* and a study staff member would contact them to inform them about the study. Those who met the eligibility criteria were subsequently invited to participate in the study.

The in-depth interview (IDI) guides were developed using a literature review to inform question and theme development (Data in S1 File). Discussions with local community partners, adolescents, and the Centre for the AIDS Programme of research in South Africa (CAPRISA) Community Research Support Group also elicited feedback on the guides. Ethical approval for the study was obtained from the University of KwaZulu-Natal Biomedical Research Ethics Committee (BREC00002517/2021). A written informed consent was completed with each participant at the start of the study, and waiver of parental consent was approved by the ethics committee for those aged 15–17 years old. Prior to the IDI, participants completed a short survey to capture basic demographic data. IDIs were completed when convenient for the participants and in their preferred language. All interviews were completed by experienced female interviewers proficient in both English and isiZulu. To avoid disruptions, interviews were conducted in a private room and lasted between 60 and 90 minutes. All interviews were conducted in isiZulu. Interviews were audio-recorded, transcribed, and then translated into English. The transcripts were quality checked against the audio recordings for accuracy by a research assistant. Participants were given ZAR100.00 reimbursement for their time and travel.

## Data analysis

This analysis used data from IDIs with 29 participants. This analysis was guided by the integrated theory of gender and power [23,24]. This theory posits that three overlapping structures govern gendered relationships between men and women: namely; (1) the sexual division of labour, (2) the sexual division of power and (3) the structure of cathexis—social norms and the affective component of relationships. These three structures are theorised as interlinked, yet distinct, and as existing on the societal and institutional levels. The theory argues that these

structures are rooted and perpetuated in society through historical, socio-political, and socio-cultural forces which seek to divide power and ascribe social norms based on gender-defined (gender-determined) roles [23,24]. The team used thematic analysis, following the process outlined by [25]. Field notes taken by the interviewer were consulted to provide additional insight into the themes identified. Initially, the research team immersed themselves in the transcripts for familiarisation with the data. Following this, two of the research team (BM & CML) developed an initial coding structure related to the overall aim of the analysis. The research team then met to review the initial codes. Additional codes emerged from the data, and the team then reviewed these codes. The full team met to discuss the themes and make any final changes and additions to the codebook and their definitions to agree on the themes and ensure saturation. Once consensus was reached, all transcripts were analysed independently by a small team (BM, CML, SM, HH), allowing for the iterative identification of themes. After coding was complete a group meeting was held to discuss the final analysis results and overall themes. The data were analysed using NVivo-12 (QSR International).

## Results

The current sample included twelve cis-gendered heterosexual males, seven cis-gendered homosexual males and ten cis-gendered heterosexual female participants (Table 1). The overall age range was 16–19 years old, and the mean age of male participants was 17.5 years old (Age range 16–19 years old), and 17.7-year-old for female participants (Age range: 17–19 years old).

The analysis identified three broad themes, 1) Interpretation and understanding of gender identity, 2) Gendered essentialism and gender roles (with sub-themes: *Young men*: *Power through providing*, and *Young women*: *The domestication process* and 3) Gender and fertility (Table 2). For the extracts reported below, please note the following conventions E = extract number, I = Interviewer, and P = Participant.

### Interpretation and understanding of gender identity

This theme deals with how young women and men in this study understood and defined gender. Throughout the interviews, gender seemed to be viewed through an essentialist lens, where biological sex and traits perceived to be inherently male, or female were key to defining gender. To identify as a specific gender entailed having certain physical attributes considered specific to and characteristic of that gender, as one male participant stated when asked about what made men different from women:

> *Eish. They are different. . . Body wise you know that a man has a penis, and a woman has a vagina.*

> (*E1:P0006 –male, 19 years old*)

When the interviewer challenged the notion of physical attributes being gender specific (with the example of facial hair) one participant offered subtle resistance highlighting the idea that the expression of gender, at least physically, is fixed:

**Table 1. Key Demographic information of the participants.**

| | Female (N = 10) | Male (N = 19) |
|---|---|---|
| Mean age | 17.7 | 17.5 |
| Age range | 17–19 years old | 16–19 years old |
| High School Education | 100% (9/9) | 100% (12/12) |

**Table 2. Outline of major themes identified during analysis.**

| Major Theme | Sub-Themes |
|---|---|
| Interpretation and understanding of gender identity | This theme deals with how young women and men in this study understood and defined gender. Gender seemed to be viewed through an essentialist lens, where biological sex and traits perceived to be inherently male, or female were key to defining gender for all adolescents regardless of sexual orientation. |
| Gendered essentialism and gender roles | Main theme deals with young women and men's understanding of gender roles in their community. The sub-theme was "Young men: Power through providing" suggests that both male and female participants perceived that a man's primary role was to be a provider, and important amongst all participants regardless of sexual orientation. |
| | The sub-theme "Young women: The domestication process" highlighted the female gender role may often be characterised by successfully engaging in domestic activities. |
| Gender and fertility | This theme dealt with the importance of bearing and fathering children for the perceived successful fulfilment of womanhood and manhood expectations by all participants. |

*P: Okay (laughing). A girl has breasts, and a boy doesn't and when a boy gets older his voice gets bigger and a girl's voice doesn't change.*

*I: Any other body parts that can help you identify or differentiate?P: A beard, only boys have it.*

*I: But I also, have it?*

*P: No K. In as much as you may say that you have it I am sure it is not the same. Yours is lighter and anyone can tell that you are a woman. As for boys it grows bigger.*

*(E2: P0011 –female, 19 years old)*

The strong association that young people reported between biological and physical characteristics and gender identity suggests they may perceive gender identity as pre-fixed and biological by nature.

## Gendered essentialism and gender roles

This theme deals with young women and men's understanding of gender roles in their community. This includes two sub-themes: 1) Young men: Power through resource provision, and 2) Young women: The domestication process.

The participants reported that, in this community, social roles were typically delegated based on gender. These roles were conceptualised by both male and female participants as predetermined and normative and linked to biological definitions of gender (as seen in Theme 1). Certain tasks, roles and responsibilities had masculine connotations while others had feminine connotations, and participants reported that gender roles restricted performing roles which were not characteristic of their gender.

*I: Are there things which you don't believe a man must do?P: I think cooking and cleaning.*

*I: Why?*

*P: I don't think that I can do that as a man.*

*I: So, you believe that they are only for women?P: Yes, I think that it is a woman's job.*

*(E3: PID 0007 –Male, 17 years old)*

Both female and male participants had similar opinions regarding division of labour and tasks, reinforced by social norms around gender expectations:

*That girls should learn to cook, clean the house and do washing. . . . They [males] must clean yards. . . .Woman is not supposed to drink alcohol.*

*(E4:P0009 –Female, 19 years old)*

The act of wanting to engage in behaviours contrary to social norms may sometimes meet with resistance. Some male participants reported enjoying cooking (a traditionally female activity) but stated that it was not allowed because it transgressed expected male behaviour:

*Oh, there is, it's cooking. I personally love cooking but cannot do it because it is considered a woman's job and at home there are a lot of women so I cannot.. . . I have tried numerous times.. . . They said I cannot cook, that there is nothing I can cook, and I must just sit down.*

*(E5:P006- Male participant, 19 years old)*

Further, the expected gender behaviours were reinforced by families and social relationships, as echoed by another young man:

*Like cooking, I really like cooking, but they don't allow me to. . .They say I must let someone who knows how to do it.*

When the interviewer followed up about other things he was prevented from doing he continued.

*Yes, there are, like cleaning the house. At home we only have one girl and I sometimes feel like she is carrying a lot and she is doing a lot on her own.. . . They say she will not be a good wife if she is failing to do all the house chores on her own.. . . Sometimes it will be elders who say that and sometimes it will be us boys who says that it is a woman's job.*

*(E6: P004- Male participant, 18 years old)*

These gender roles may serve to enhance traditional gender norms premised on the natural expression of masculinity or femininity. These are discussed in the two sub-themes below:

*Young men*: *Power through resource provision.*

This sub-theme suggests that both male and female participants perceived that a man's primary role was to be a provider, in spite of the fact that a large number of their households were headed by women. The provider role has connotations of strength and power, typically afforded to men, and embodies the idea that men, by virtue of their manhood, were naturally suited as the heads of the household (Mabaso *et al.*, 2018). As one participant reports:

*You must be the head of the house and be a bread winner. . . .You must take care of the family and protect the household (E7: P0004 –Male, 18 years old)*

Another male participant notes that men derive a sense of manhood from being providers and may subsequently feel devalued and emasculated if they are unable to fulfil the provider role or if their spouse is 'more' of a provider:

*When you grow up you are expected to provide for your family, it doesn't look okay when you are not working, and your wife is, [that is] when life gets difficult. Also, when you are earning less than your wife it becomes difficult and that normally wouldn't sit right with you as a man.* (E8:P0003 –Male, 17 years old)

The above quote also suggests that while men may be comfortable with both partners contributing towards the household, if a female partner becomes the perceived primary provider it may be construed as a transgression of traditional gender norms. The male participants spoke of the provider role being an expectation of being a man in their communities, giving a sense that this may have been a role they were forced to take regardless of whether they wanted this responsibility. Interestingly, many of the female participants felt that women were expected to take on multiple roles, including going to work but that this did not appear to result in them gaining power. Instead, women are expected to contribute to the household through work, while also fulfilling household roles and observing traditional gender norms:

*Being a woman is very hard. . . You have to be the one who carries babies in your belly, you have to be the ones that makes sure that you have cooked food to be eaten, you have to take care of children and people. In some cases, you as a woman works and you still need to come back and make sure that everything in the house remains in order. It is very hard. A man can go to work and come back and ask for food from a woman and not do it themselves.*

*(E9: P0011 –Female, 19 years old)*

The importance of the provider role may be related to the accrual of power which comes with it *for men*. Most participants felt the provider role was synonymous with being the head of the household, a position typically afforded to men, and imbued with social power and control over of the family [26]. Situated within social contexts that remain strongly patriarchal could explain why men are reluctant to forfeit their control, and why dissonance and the maintenance of traditional gender norms may arise in situations where women contribute to the household [26]. Thus, it may be that men take on the role of ensuring resources come into the household, regardless of who earns these, creating a "provider role" that affords men more power over women and families even if they are not the sole provider. In comparison, and aligning with the binary theme, women are often delegated roles within the household which are predicated on subservience to their male counterparts even in situations where they may fulfil a provider role or co-provider role.

*Young women: The domestication process*

All participants reported that domestic chores were unquestionably assigned to young girls based on their gender with little deviation from this, normalising the idea that domestication is a condition of womanhood.

*It[s] difficult to be female. Men do not clean; they do not cook they just eat and leave everything there and female(s) must stay in the house [and] cook if you have to and clean the house.*

*(E10:P0002 –Female, 18 years old)*

Interestingly, a male participant (see Extract 6) expressed that he has attempted to equalise the division of labour within his household by helping his female sibling with household

chores. However, rather than being viewed as helpful he is viewed as a hindrance by elders, demonstrating how the feminisation of domestic labour may be generationally enforced. Domestic chores being contingent on gender may promote the idea that gender roles are binary. Importantly, this may be enforced intergenerationally demonstrating how context impacts normative roles.

The importance of domestication is premised on the idea that it is a fundamental condition of womanhood as it invariably helps women to fulfil future womanly roles (such as being a wife and a mother). A participant expanded on this by explaining how young girls are introduced to domestic labour as a sort of quasi-initiation into womanhood:

*Yes, ahh they [girls] are taught house chores for future purposes for when they get married and have their own houses and children.*

*(E11:P0003 –Male, 17 years old)*

Domestication often has connotations of subservience. By entrenching the idea that domestication is what makes a 'good woman' from a young age, young girls may be incorporating subservience into their conceptions of womanhood. This may subsequently solidify their place in heterosexual relationships as less powerful and ultimately generate dependence on their male counterparts even if they are unhappy with this, as reported below:

*When you are a woman, you are always supposed to be humble, even when you are maybe having an argument with your partner you are expected to be quiet and listen to him to speak. We however struggle with it*

*(E12:P0005 –Female, 18 years old).*

Another participant notes the subservient role which women are expected to fulfil in her community:

*Women should respect men and [a] married woman must treat her husband like a king.*

*(E13: P0002 –Female, 18 years old).*

The findings suggest that gender roles, much like conceptions of gender identity, may be viewed as prefixed and socially sanctioned. Transgressing gender roles may generate feelings of uneasiness precisely because certain activities are constructed as incompatible with contextual ideas of masculinity and femininity. Thus, fulfilling a gender role (whether it be the male or female) becomes analogous to being a man or woman. Importantly, the entrenchment of traditional gender roles that are socially sanctioned enhance power disparities between young men and women.

## Importance of fertility for successful womanhood and manhood

Bearing and fathering children was perceived as a key measure of successful womanhood and manhood by all participants. The female participants reported that fertility was highly prized in the community and that successful womanhood entailed having children. This was highlighted by one participant when asked about how she identifies herself as a woman:

*Because of things that are happening in my life. Like how my body develops points out that I am a woman. I also proved that I am a woman because. . .because I was able to fall pregnant and have a baby. I am actually a mother now.*

*(E14: P0005 –female, 18 years old)*

Social pressure placed on young women to have children may highlight a tension between parental and adolescent SRH concerns and perceived social expectations driving teenage pregnancy also referred to in this paper as early pregnancy. Participants reported that this perceived social pressure made childbearing feel like a social expectation that women needed to fulfil. Women who chose not to have children, or were infertile, were subsequently chastised and coerced by the wider community. Age was an important factor mediating how childbearing affected the definition of a woman's success in her traditional role. The pressure to have a child increased as a woman got older–resulting in negative consequences should she fail:

*Some get insulted and called names like "inyumba". Your wellbeing ends up affected because of the hostile environment. Even when you get into a fight with someone, or an argument, [and they] know that you are sexually active, they will call you "inyumba".*

*(E15*: *P0005 –Female, 18 years old).*

The frequently reported pejorative ('inyumba') directed at women without children suggests that fertility is a gendered phenomenon. This word loosely translates to 'barren' which harbours connotations that more aptly apply to infertile women rather than impotent men. The use of this term thus seems to suggest that infertility (or a lack of children) is conceptualised as a barrier to womanhood, or the fault of the woman. One participant reports that some women who don't fulfil the mandates of fertility are even exposed to abuse:

*I think they [women who don't have children] are treated badly. Men beat up their wives and becomes abusive because they [women] cannot have children.*

*(E16:P0013 –Male, 16 years old)*

While communities may express concern over the risks associated with adolescent sex, fear of infertility, social pressure, social expectations, and the negative consequences of not having children, may inadvertently push adolescent girls to have children earlier. This may be done to avoid possible negative consequences and make them feel rewarded for successfully proving their fertility. One participant demonstrates the fear of infertility:

*I will feel really bad, and I think I will reach a point where I will feel suicidal because it would be really hard to accept that.*

*(E17:P0014 –Male, 17 years old)*

Another young woman also noted the social pressure:

*When I look at things now, we have children as young as we are. Before it used to be a disgrace to have a child as young as we are but now [people are] acting like it's a competition. We [people in my community] don't see it as a disgrace anymore and the elders have now accepted that it's something you are going to do. It's so common.*

*(E18:P0005 –Female, 18 years old)*

Thus, contextual, and social factors may serve to create an environment in which adolescent girls feel pressure to have children. At a structural level, gender and cultural ideals may

intersect to pressure adolescent girls to prove their fertility. Participants felt that childbearing was an inherent part of their culture which they were meant to uphold, as illustrated in response to a question about what Zulu women feel obligated to do in relationships:

> [*I must*] *give birth.. . .Yes, that one is non-negotiable if you are a woman, it must happen.*
>
> *(E19:P0009 –Female, 19 years old)*

For the male participants, a key issue appeared to be concerns over the continuation of lineage and name. When discussing lack of paternity or an absence of children a participant notes the implications of not having children on lineage in his culture and the part it may play:

> *Those people are normally mistreated, and they get called names, they say you are barren* (*inyumba*), *and they talk down on you and say when you die no one will remain behind and carry your name.*
>
> *(E20:P0003 –Male, 17 years old)*

The same participant goes on to explain that in his culture the prizing of fertility is also related to producing a male heir, or provider, for the future generation:

> *. . ..What can I say*? *In every household there needs to be a man, no matter how many female children there are, a male child is needed as an heir to the family and that heir must be a man.. . . Let me just say in Zulu culture an heir needs to be and must be a male child.*
>
> *(E21:P0003 –Male, 17 years old).*

Similar to female participants, these concerns may impact on the behaviour of young men, driving and placing pressure on young men to father children earlier. Thus, the social pressure and expectations that are associated with proving fertility for young women and men may have backfire effects for SRH outcomes, especially pregnancy. For example, the idea that one needs to prove their womanhood through childbirth may consequently motivate adolescent women to have children in order to avoid being chastised and belittled within their community. For men, having a child becomes a sign of virility, and a means of safeguarding his lineage. The findings highlight how gender roles and social expectations may influence sexual decision making and how the social push for bearing children may be in conflict with messaging around the possible negative health outcomes of adolescent pregnancy. Importantly, the findings suggest that it may be possible that some SRH outcomes offer a means of achieving perceived social expectations and avoiding possible social sanctioning.

## Discussion

This analysis sought to understand gender roles and norms amongst adolescents living in rural KwaZulu-Natal and how they may influence sexual and reproductive health outcomes and decision-making. Findings suggest that individual autonomy can sometimes be challenged by gender norms which influences how adolescents behave, what attitudes they may develop and the decisions they make. Within this community, a context has been created which prioritises and enforces specific definitions of gender roles. Roles which are transgressed (i.e., wanting to engage in household behaviours that are not normative for your sex), or go unfulfilled (i.e., producing children), typically come with social consequences. Our study findings demonstrate that gender roles and norms may influence multiple realms of life, including the sexual realm,

where they serve a central function in maintaining traditional power structures, and maintaining male dominance within relationships, family structures and decision-making processes. The maintenance of these unequal power dynamics, endorsed by socially sanctioned gender and social norms, may make independent SRH decision-making more difficult for women. This may increase their risk of sexual and gender-based or intimate partner violence, while entrenching negative gendered expectations that keep women dependant on their partners/ families and sustain asymmetrical power relations.

Results from this study underscore how gender roles are derived from broader understandings of gender identity and gender norms. Within this sample, community adolescents conceptualised gender as a binary. Men and women were perceived as diametrically opposed which stimulated the idea that men and women should fulfil differing, and often opposing, gender roles. The common use of biological reference points in the context of gender identity and gender roles naturalised the idea that there are, by nature, a fixed set of gender roles which are predetermined and intrinsic to manhood or womanhood. This in essence entrenches the notion that certain gender roles are incontestable precisely because they appear 'natural'. Surprisingly, although participants in this study were all born after the year 2000, and included homosexual males, they all held varying degrees of traditionalist views when it came to gender roles. This could indicate that gender roles are being passed on intergenerationally (such as by elders in the community) quite early on as they appear static and fixed. Further, it suggests that the dominant definitions of gender and social norms align with more conservative definitions, and seem prevalent within our samples experience, across different age ranges. Some results suggest that there may be some 'shifts' from culturally held views. Some male adolescents showed an interest in activities such as cooking (considered female roles). This may suggest that young men are open to expanding their definitions of the male role and highlights the need for interventions amongst younger men, changing cultural views over time as these men get older. Additionally, we saw an anomaly with regards to a women's role as household providers. Even if they played a provider role they are still perceived as subservient to men, suggesting that economic empowerment interventions aimed at women may not be sufficient to led to shifts in gender roles, and norm and that gender equality interventions should be part of all social and health interventions.

Amongst study participants young men were expected to incorporate a provider identity into their concepts of manhood. This entailed having material resources, safeguarding the family and being the head of the household, all of which had connotations of power. These findings are consistent with other findings in South Africa [20,27–29]. Further, even if women fulfilled a provider role by bringing in resources, our findings suggest that this may not imbue women with power. Rather men remain in control of the household, suggesting that the provider role is synonymous with maleness and the management of resources rather than providing the resources themselves. Importantly, in the context of sexual health, literature has shown that the unequal control of resources by men (which is instilled from a young age) contributes to the development of broader unequal power relations between men and women [1]. These unequal relations could in turn have a negative effect on the negotiation of safer sex in later adulthood. Indeed, studies indicate that unequal economic power restricts sexual autonomy and results in coercive sexual practices which increase the risk of negative SRH outcomes [13].

Contrastingly, young women were expected to fulfil a caregiver role and undergo a domestication process during adolescence. This role was presented as intrinsic to womanhood, akin to a normative biological process, which prepared adolescent girls for adulthood. We anticipated seeing shifts in the definition of female and male gender norms amongst our younger sample, especially considering they have grown up in the digital and social media age. However, our findings were congruent with older South Africa studies [30–33] which highlight the

importance of the "caregiver role" of women. Some of these studies posit that this role serves a disempowering function for women which elevates risks of sexual and gender based or intimate partner violence and economic dependency. Additionally, it has been reported that women in unequal relationships are exposed to greater SRH risks, including HIV. Conversely, research shows that gender attitudes which emphasise egalitarian relationship dynamics (rather than power and submission) are related to better sexual communication and higher usage of contraception [30–33].

Study results also illustrate how gender roles in this context are intertwined with fertility expectations and, notably, how these were largely aimed at young women. It was reported in the interviews that to be a woman entailed having children, which could possibly influence the individual decision making around wanting children [34]. Those who were unable to bear children, or perhaps did not want children, succumbed to social consequences. Much like the above-mentioned gender roles, the mandate of fertility was grounded in dominant gender and social norms and seemingly non-negotiable expectation which women were meant to uphold on the premise that bearing children was a natural extension of womanhood [34,35]. A tangible benefit of having children (for young women) could be the avoidance of negative social consequences through the affirmation of womanhood and this may have the unintended consequence of promoting early pregnancy amongst adolescent girls. It could also come with the consequence of assuming childcaring responsibility in lieu of factors such as the pursual of education. These areas require future research.

Conversely, for young men, paternity acted as an affirmation of manhood through the maintenance of power. Foremost, having children serves as a marker that a man is virile, an important aspect of manhood [13,16]. Additionally, having children, irrespective of their gender, also has tangible benefits for men in terms of power maintenance. For example, having a male child ensures that a man may pass on his lineage and ensures that his lineage maintains power over the household while having a female child ensures that caregiver responsibilities in the household are still maintained. These gender roles, which are grounded in gender and social norms related to affirming one's manhood or womanhood, may subsequently increase the risk of early pregnancy and expose adolescents to sexual health vulnerabilities.

Interventions aimed at shifting harmful gender norms amongst early adolescents and the broader community are needed to promote positive gender and social norm development in the present while promoting long-term shifts in gender norms as adolescents gets older. Programmes and interventions, like SASA!, Stepping Stones [36] and DREAMS [37,38] offer important lessons that comprehensive gender based interventions can promote healthier gender role development and lead to positive change in male and female relationships. Information based interventions like life orientation and comprehensive sexual health intervention also offer means of fulfilling information gaps. . . Indeed, our participants who had already attempted to transgress these gender roles alluded to the difficulties they had faced from elders and broader systems in their community. Promoting positive gender norm changes amongst adolescents and adults can lead to structural shifts in the definitions of personhood and promote changes in how communities think about healthy adolescent identity development and sexuality. This broader community change is needed for shifting existing supportive policies, and the availability of adolescent SRH interventions to actual real-world use by adolescents in an environment that supports their autonomy to make sexual decisions.

Our study has some limitations. Firstly, the sample size is relatively small, but aimed to achieve representation of sex, and age range. We oversampled for men which is a group often underrepresented in sexual health research. Secondly, some of the male participants identified as homosexual but were not asked how their sexual orientation impacted their vision of gender roles and how they played out in homosexual relationships, if at all. Finally, participants all

came from a specific rural geographic location. This means we may have a more homogenous sample in terms of language, traditional/social influence and strongly held community norms. However, the findings still have possible transferability to similar contexts, and as the community is not isolated from close by larger urban centres it is possible that adolescent's perceptions are informed by information and views beyond their immediate context. There is a possibility that we achieve limited representation because the study specifically engaged adolescents who were willing to disclose their sexual activity, may represent adolescents who become sexually active at an earlier age than others (by choice or not) and the snowballing sampling may have meant we overrecruited young people in similar peer networks. Future research is, however, needed which compares these findings with a sample of adolescents from differing social backgrounds, such as in urban and different wealth contexts.

## Conclusions

The results of this study highlight that adolescent behaviour operates within the context of community gender norms. Rather than solely targeting individual behaviours, SRH campaigns need to target individual and community gender norms amongst both young adolescent and adult groups. This way the focus is not only on changing norms in the short to medium term, but making positive normative changes that are sustained and infused into community belief-systems as the adolescents grow up and become adults. This can be accomplished through gender-based interventions and the use of workshops which discuss gender-role development and address the link between harmful gender norms, negative sexual health outcomes and other associated social harms. These workshops could be held across varying demographic levels to ensure long term sustained change. It is hoped that by actively questioning existing gender stereotypes, deconstructing gender binaries, and encouraging egalitarian relationships across multiple levels that harmful gendered expectations can likewise be dismantled. Importantly, these interventions should promote the idea of gender equity holistically and reformulate ideas pertaining to what a successful 'woman' or 'man' entails. This in turn could purposefully challenge longstanding power inequalities between men and women and, importantly, ensure that young women are no longer confined to a role of dependency and that young men are allowed to grow beyond the boundaries of toxic expectations of gender roles.

## Supporting information

**S1 File. Qualitative data collection guides for interviews.**
(PDF)

## Acknowledgments

The study team would like to extend their utmost gratitude to the study participants and the CAPRISA Vulindlela research team.

## Author Contributions

**Conceptualization:** Brett Marshall, Lucia Knight, Hilton Humphries.

**Data curation:** Hilton Humphries.

**Formal analysis:** Brett Marshall, Celia Mehou-Loko, Sindisiwe Mazibuko, Lucia Knight, Hilton Humphries.

**Funding acquisition:** Hilton Humphries.

**Methodology:** Lucia Knight, Hilton Humphries.

**Project administration:** Brett Marshall, Makhosazana Madladla.

**Supervision:** Hilton Humphries.

**Validation:** Hilton Humphries.

**Visualization:** Hilton Humphries.

**Writing – original draft:** Brett Marshall, Celia Mehou-Loko, Sindisiwe Mazibuko, Makhosazana Madladla, Lucia Knight, Hilton Humphries.

**Writing – review & editing:** Brett Marshall, Celia Mehou-Loko, Sindisiwe Mazibuko, Makhosazana Madladla, Lucia Knight, Hilton Humphries.

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
