## [Decision Letter · Decision Letter 0]

5 Jul 2023

PONE-D-23-11309Exploring perceptions of gender roles amongst sexually active adolescents in rural KwaZulu-Natal, South AfricaPLOS ONE

Dear Dr. Humphries,

Thank you for submitting your manuscript to PLOS ONE. After careful consideration, we feel that it has merit but does not fully meet PLOS ONE’s publication criteria as it currently stands. Therefore, we invite you to submit a revised version of the manuscript that addresses the points raised during the review process.

We look forward to receiving your revised manuscript.

Kind regards,

Obasanjo Afolabi Bolarinwa, Masters

Academic Editor

PLOS ONE

Journal Requirements:

"This work was supported by the South Africa National Research Foundation’s Thuthuka Funding under Grant [number TTK200403511178]."

"This work was supported by the South Africa National Research Foundation’s Thuthuka Funding under Grant [number TTK200403511178]."

Reviewers' comments:

Reviewer's Responses to Questions

**Comments to the Author**

1. Is the manuscript technically sound, and do the data support the conclusions?

Reviewer #1: Yes

Reviewer #2: Yes

2. Has the statistical analysis been performed appropriately and rigorously? 

Reviewer #1: N/A

Reviewer #2: Yes

3. Have the authors made all data underlying the findings in their manuscript fully available?

Reviewer #1: Yes

Reviewer #2: No

4. Is the manuscript presented in an intelligible fashion and written in standard English?

Reviewer #1: Yes

Reviewer #2: Yes

5. Review Comments to the Author

Reviewer #1: The manuscript is well written, and is technically sound. The manuscript addresses an important topic on gender roles vis a vis adolescents’ sexual reproductive health among adolescents in South Africa. It will be strengthened by addressing the following minor comments. See attached report.

Reviewer #2: General

Given the gender disparity in HIV and sexual risk from an early age in South Africa, this research has a valuable goal - to achieve a detailed understanding of gender roles among adolescents. With gender disparities in HIV risk among 15-19 year olds greater than any other age group, it is valuable that the study explored this topic with 16-19 year olds. The paper is well written and presented, and offers valuable insights (with compelling quotes from young people) to strengthen programming for healthy and equitable youth development in KwaZulu-Natal. In a number of ways, it could be strengthened by deepening the enquiry and rigour, and exploring more diversity among young people’s perceptions and experiences.

Introduction

Paragraph 2 appears to link adverse sexual outcomes with young women’s behaviours and decisions. For example, the sentence in Lines 61-64 appears to attribute experiences of violence to women’s ability to make decisions (when, by definition, acts of violence are taken by force). And the terms ‘risk-taking’ and ‘risky sexual behaviours’ imply individual agency or choice, with an implication of irresponsibility. The relationships (conceptual) between social norms, individual agency, and health and violence risks could be made more clear.

It would be helpful to distinguish social and gender norms, as they seem to be used interchangeably. Please also define ‘gender success’ and was it defined from young people’s perspectives in this research?

Also, the terms “positive” and “negative” are used frequently to describe gender and social norms, but have not been defined. Which values or whose values are used to make these descriptions?

Methods

There is little detail about the social context of the study setting, or why it was chosen. What are the schooling, marriage, ethnic and religion profiles in Vulindlela, to help interpret the findings as well as their applicability elsewhere? And to understand if the integrated theory of gender and power has validity in this socio-cultural context.

Study Sample: some aspects of the study sample are unclear, for example:

- The participants are sometimes described as 15-19 years (in lines 108 and 161) and other times aged 16-19. In lines 161-163, the age range is described as 15-19, yet neither age range for males nor females includes age 15.

- In line 110: it’s not clear what recall “bias” would be at play or how it would distort the results (differential recall among those with adverse sexual outcomes compared to those without such outcomes?)

- It seems more likely that there is selection bias (or limited representation), in that:

- (1) those willing to disclose their sexual activity (in screening questions) and participate were different to other adolescents unwilling to disclose this private information? Particularly if there are social expectations about teenage sexuality activity, and those expectations differ among males and females. (Is this why there were more male than female as participants? 19 vs 10)

- (2) the 29 participants may represent adolescents who become sexually active at an earlier age than others (by choice or not). What is the mean age of first sex in this context?

- Also, snowballing may have recruited young people in similar peer networks (extending the sampling bias) rather than expanding the diversity of adolescents.

- The Discussion states that you “sampled for representativity” – it wasn’t clear in the Methods how this was done, or what kind of representation was sought.

Data and Tools

The data are described as “longitudinal”, with data from 2 interview rounds used for this analysis. Is the analysis treated as longitudinal, to identify changes with age or time? How were data analysed for those with only 1 interview, if lost to follow up, versus participants with 2 interviews?

Can the interview guides be included in supplementary material? Was the same guide used in both interviews?

Did the authors conduct a reflexivity statement to explore their own active role in the results generation? Can this be included in the submission material?

Related to this point, Braun & Clarke 2006 (the methodology cited) are critical of describing analysis in ways such as “additional codes emerged from the data” (line 150) since describing the analysis as a passive process denies the active role of the researchers.

Also, is ‘inter-rater reliability’ the goal in an analysis process that is inherently subjective?

How did the researchers determine when saturation was achieved?

Can the authors justify use of the integrated theory of gender and power, for analysis, rather than an inductive or grounded theory approach?

Results

Can a coding tree be included with the submission, for more detail about the data and themes identified by the researchers?

It was specified in Methods that both heterosexual and homosexual males were recruited, however, there is no distinction made by sexual orientation in the Results. Also, the findings seem to apply only to heterosexual relationships: is that the case? Should we conclude that gender norms and expectations (of heterosexual relations) are the same regardless of male sexual orientation, or was this not explored in the analysis? This seems relevant to an analysis of sexual relationship power, and an opportunity for learning from the diverse experiences of adolescents.

Could use of the theory of gender and power have led to findings that supported the theory, with its strong reliance on traditional gender roles and patriarchal influences? Thus impeding the analysis from finding more variation and complexity – a possibility highlighted in this relevant manuscript: https://www.ncbi.nlm.nih.gov/pmc/articles/PMC7170748/pdf/nihms-1540698.pdf

May it also have impeded learning from both heterosexual and homosexual males?

(Also, shouldn’t the Connell paper from 1987 be used as the primary reference for this theory?)

It is difficult to interpret statements on ‘early’ pregnancy or childbearing without understanding what qualifies as early. From when (what age) is childbearing a social expectation for young women? And by what age would a young male like P0014 feel so bad if he has not demonstrated fertility? Also, are there social circumstances that are expected along with childbearing, e.g., the context of marriage or partnership, or economic stability? E18 suggests that expectations about early childbearing have changed in this community – why may social stigma about teen pregnancy (previously “a disgrace”) have changed? This also suggests it may not be embedded in tradition or social norms, if expectations of early pregnancy have recently increased.

Discussion

In the Discussion, links between gender roles and HIV / SRH outcomes are made, based on assumptions or other studies. Did the research explore these links more explicitly? E.g., how adolescents perceive and navigate tensions between pressures to prove fertility and prevent HIV/STI infection? (Again, it would help to see the Topic Guide to understand what was asked)

As noted, early pregnancy, here and throughout, is presented as an adverse outcome but what is considered “early” pregnancy in this social context? (and in relation to the legal age of consent) And in terms of young women’s preferences and choice?

Lines 527-530 call for interventions to shift harmful gender norms among early adolescents, and the existing evidence base from numerous community-based trials, including SASA! and Stepping Stones and MTV Shuga, among other campaigns, which could have been cited for relevance and lessons. Also, comprehensive school-based health education (UNESCO-led curricula adopted by many countries) addresses this to an extent, and could have been considered here too.

Other research finds that power dynamics in heterosexual relationships are exacerbated by age disparities between females and older partners, but this isn't addressed in this analysis or in the interpretation. In such circumstances, focusing interventions on early adolescents or even older adolescents will have limited impact. (And programmes like Stepping Stones may have more impact, as they include adult men)

Given the number (only 10 females) and ways that the participants were sampled, it overstates the findings to say that social norms (demonstrated by the participants) are “pervasive” or across different age groups. (line 472) For reasons listed above, there may be more diversity and variation that was not identified due to the way the sampling and analysis was conducted, and the choice of theoretical framework.

6. PLOS authors have the option to publish the peer review history of their article (what does this mean?). If published, this will include your full peer review and any attached files.

Reviewer #1: **Yes: **Busisiwe Nkosi

Reviewer #2: No

---

## [Author Response · Author response to Decision Letter 0]

25 Aug 2023

A full tabulated response to queries has been included as a file with the submission. Please let us know if this is not sufficient.

---

## [Decision Letter · Decision Letter 1]

19 Dec 2023

Exploring perceptions of gender roles amongst sexually active adolescents in rural KwaZulu-Natal, South Africa

PONE-D-23-11309R1

Dear Dr Humphries,

We’re pleased to inform you that your manuscript has been judged scientifically suitable for publication and will be formally accepted for publication once it meets all outstanding technical requirements.

Kind regards,

Obasanjo Afolabi Bolarinwa, Masters

Academic Editor

PLOS ONE

Additional Editor Comments (optional):

Reviewers' comments:

Reviewer's Responses to Questions

**Comments to the Author**

1. If the authors have adequately addressed your comments raised in a previous round of review and you feel that this manuscript is now acceptable for publication, you may indicate that here to bypass the “Comments to the Author” section, enter your conflict of interest statement in the “Confidential to Editor” section, and submit your "Accept" recommendation.

Reviewer #2: All comments have been addressed

Reviewer #3: All comments have been addressed

2. Is the manuscript technically sound, and do the data support the conclusions?

Reviewer #2: Partly

Reviewer #3: Yes

3. Has the statistical analysis been performed appropriately and rigorously? 

Reviewer #2: N/A

Reviewer #3: Yes

4. Have the authors made all data underlying the findings in their manuscript fully available?

Reviewer #2: No

Reviewer #3: Yes

5. Is the manuscript presented in an intelligible fashion and written in standard English?

Reviewer #2: Yes

Reviewer #3: Yes

6. Review Comments to the Author

Reviewer #2: (No Response)

Reviewer #3: The study explored the perception of gender roles among adolescents in SA and how this influences sexual and reproductive health. Gender norms and cisgender adolescent were inclusive in the study.

Find below few observation:

There is need for clarity of below terms such as gender inequalities, unequal power dynamics, harmful gender, gender norms, and adolescent sexuality norms

Methods

how were participants approached for the study

could you make the sampling more explicit and systematic to follow through

How was sample size determined

How many people rejected to participate in the study

Where precisely was the data collected e.g home or farm

Was there anyone present during the IDI beside the respondents

Was video or audio or note used for recording information?

How was theme arrived at for the results

Discussion

Please integrate previous studies and gaps in your discussion

What is the limitation of the study

What is the contribution to knowledge from this study

7. PLOS authors have the option to publish the peer review history of their article (what does this mean?). If published, this will include your full peer review and any attached files.

Reviewer #2: No

Reviewer #3: No

---

## [Editor Report · Acceptance letter]

5 Jan 2024

PONE-D-23-11309R1 

PLOS ONE

Dear Dr. Humphries, 

I'm pleased to inform you that your manuscript has been deemed suitable for publication in PLOS ONE. Congratulations! Your manuscript is now being handed over to our production team.

Kind regards, 

on behalf of

Dr. Obasanjo Afolabi Bolarinwa 

Academic Editor

PLOS ONE